# Metabolic and Structural Insights into Hydrogen Sulfide Mis-Regulation in *Enterococcus faecalis*

**DOI:** 10.3390/antiox11081607

**Published:** 2022-08-19

**Authors:** Brenna J. C. Walsh, Sofia Soares Costa, Katherine A. Edmonds, Jonathan C. Trinidad, Federico M. Issoglio, José A. Brito, David P. Giedroc

**Affiliations:** 1Department of Chemistry, Indiana University, Bloomington, IN 47405-7102, USA; 2Instituto de Tecnologia Química e Biológica António Xavier, Universidade Nova de Lisboa, 2780-157 Oeiras, Portugal; 3Instituto de Química Biológica de la Facultad de Ciencias Exactas y Naturales (IQUIBICEN)-CONICET and Departamento de Química Biológica, Universidad de Buenos Aires, Buenos Aires C1428EHA, Argentina; 4Department of Molecular and Cellular Biochemistry, Indiana University, Bloomington, IN 47405-7003, USA

**Keywords:** hydrogen sulfide toxicity, reactive sulfur species, persulfide, persulfidation profiling, coenzyme A persulfide, X-ray structure, fatty acids

## Abstract

Hydrogen sulfide (H_2_S) is implicated as a cytoprotective agent that bacteria employ in response to host-induced stressors, such as oxidative stress and antibiotics. The physiological benefits often attributed to H_2_S, however, are likely a result of downstream, more oxidized forms of sulfur, collectively termed reactive sulfur species (RSS) and including the organic persulfide (RSSH). Here, we investigated the metabolic response of the commensal gut microorganism *Enterococcus faecalis* to exogenous Na_2_S as a proxy for H_2_S/RSS toxicity. We found that exogenous sulfide increases protein abundance for enzymes responsible for the biosynthesis of coenzyme A (CoA). Proteome *S*-sulfuration (persulfidation), a posttranslational modification implicated in H_2_S signal transduction, is also widespread in this organism and is significantly elevated by exogenous sulfide in CstR, the RSS sensor, coenzyme A persulfide (CoASSH) reductase (CoAPR) and enzymes associated with de novo fatty acid biosynthesis and acetyl-CoA synthesis. Exogenous sulfide significantly impacts the speciation of fatty acids as well as cellular concentrations of acetyl-CoA, suggesting that protein persulfidation may impact flux through these pathways. Indeed, CoASSH is an inhibitor of *E. faecalis* phosphotransacetylase (Pta), suggesting that an important metabolic consequence of increased levels of H_2_S/RSS may be over-persulfidation of this key metabolite, which, in turn, inhibits CoA and acyl-CoA-utilizing enzymes. Our 2.05 Å crystallographic structure of CoA-bound CoAPR provides new structural insights into CoASSH clearance in *E. faecalis*.

## 1. Introduction

Recent studies based on seminal work [1] suggest that H_2_S may be a clinically relevant bacterial adaptive response to host-derived oxidative stress and antibiotics during infections [2,3,4,5,6,7]. These cytoprotective properties attributed to H_2_S may well be the result of downstream, more oxidized sulfur species, collectively termed reactive sulfur species (RSS) [8]. One of these RSS is the organic persulfide (RSSH), which has enhanced nucleophilicity compared to its thiol counterpart and readily reacts with infection-relevant oxidants and β-lactam antibiotics [8,9,10,11]. Although much of the work on RSS has focused on mammalian systems [8], RSS biogenesis has been shown to be present in a number of bacteria [7,12,13,14].

The beneficial impact of H_2_S and RSS can only be accessed when the negative impact of sulfide poisoning [15] and over-persulfidation of the thiol metabolome and proteome can be limited [7,16] in a process we call H_2_S/RSS homeostasis. Maintaining H_2_S/RSS homeostasis may be particularly important in the sulfide-rich environment of the gastrointestinal tract, where H_2_S has been documented to be physiologically beneficial [17,18,19] and linked to several gut-derived human diseases at elevated concentrations [20,21,22]. Sulfide has been estimated to range from ~0.2 to 2.4 mM in this niche derived from both host- and microbially-derived sulfate reduction [23]. Furthermore, a recent study suggested that taurine, an organosulfonate produced by the host during infections, enables the host to resist colonization by pathogens by increasing endogenous H_2_S production from taurine by resident microbiota [24]. This suggests that organisms, both commensals and pathogens, that inhabit this niche must evolve ways to sense and detoxify H_2_S and consequential downstream oxidized sulfur species [25].

All bacteria encode RSS-sensing transcriptional regulators that control the expression of genes that encode enzymes that are known or projected to be involved in the biogenesis and clearance of H_2_S/RSS [25]. Previously, we identified and characterized the *cst*-like operon in the human pathogen and commensal member of the gut microbiota *Enterococcus faecalis* [12]. This operon is transcriptionally regulated by the RSS-sensing transcriptional repressor CstR [26,27,28]. The *cst*-like operon consists of two sulfurtransferases, RhdA and RhdB, which possess thiosulfate sulfurtransferase (TST) activity; however, their in-cell sulfur donors and acceptors are not yet known. Additionally, CstR regulates *coaP*, which encodes the coenzyme A (CoA) persulfide (CoASSH) reductase (CoAPR), which reduces CoASSH to the free thiol and H_2_S [12]. The biogenesis of H_2_S by CoAPR combined with its specificity toward CoASSH over other low molecular weight (LMW) persulfides implies that RSS, specifically CoASSH, confers a greater physiological impact relative to H_2_S in this organism. The fate of CoAPR-generated H_2_S is not known, nor is the mechanism by which *E. faecalis* generates LMW persulfides, although these species are readily detected and quantitated in soluble cell lysates [12].

Here, we sought to investigate the physiological response of *E. faecalis* to conditions of sulfide mis-regulation by adding exogenous Na_2_S as a probe of a pathophysiological response to elevated H_2_S. A proteomic analysis revealed an increased cellular abundance of CoA- and acyl-CoA requiring enzymes, including CoAPR, with the latter anticipated on the basis of regulation by CstR [12]. Protein *S*-sulfuration (persulfidation) profiling reveals that ≈13% of the proteome is persulfidated and identifies potential regulatory targets of H_2_S/RSS signaling, including enzymes involved in de novo fatty acid biosynthesis and the acetyl-CoA-producing enzyme phosphotransacetylase (Pta). As anticipated, both CstR and CoAPR are heavily persulfidated in cells consistent with their proposed mechanisms of action [12,27,28]. In addition, the spectrum of fatty acids changes significantly following the addition of exogenous sulfide; this is coupled with a significant increase in cellular acetyl-CoA, the major cellular fate of which is fatty acid biosynthesis. We demonstrate that CoASSH is a potent inhibitor of *E. faecalis* Pta, which suggests that CoASSH poisoning of CoA- and acyl-CoA-requiring enzymes [29] may underscore the metabolic burden of H_2_S toxicity in this organism. Finally, our crystallographic structure and molecular dynamics simulations of CoA-bound *E. faecalis* CoAPR provide new mechanistic insights into CoASSH clearance.

## 2. Materials and Methods

### 2.1. Growth of Enterococcus faecalis

Unless otherwise noted, all cultures were grown microaerophilically, defined here as static cultures in sealed culture tubes at 37 °C with minimal head-space. *E. faecalis* strain OG1RF (wild-type, WT) was grown overnight in tryptic soy broth (TSB) medium, supplemented with 0.25% (*w/v*) glucose. Overnight cultures were diluted into fresh medium at an OD_600_ ≈ 0.007 and grown until ≈0.1 as a preculture. Precultures were then diluted into fresh medium to an OD_600_ of ≈0.007 and grown until OD_600_ ≈ 0.2, at which time cells were treated as described for each experiment below.

### 2.2. Proteomic Analysis of Wild Type E. faecalis before and after the Addition of Exogenous Na_2_S

Quadruplicate 15 mL cultures of wild type *E. faecalis* were collected by centrifugation for untreated cells or treated with 0.2 mM Na_2_S for 30 min followed by centrifugation. All cells were washed with ice-cold PBS and stored at −80 °C. Pellets were thawed on ice and resuspended in 600 μL lysis buffer (25 mM HEPES, pH 7.4, 150 mM NaCl, 5 mM tris(2-carboxyethyl) phosphine hydrochloride (TCEP), EDTA-free protease-inhibitor cocktail [1:500] dilution). Resuspended cells were transferred to Lysing matrix B tubes and lysed in a bead beater with a rate of 6 m/s for 45 s, three times, with 5 min cooling on ice in between runs. Samples were centrifuged at 13,200 rpm for 15 min at 4 °C. The supernatant was transferred to a new 1.5 mL tube and the protein concentration was quantified by a Bradford assay using a standard protocol. A total of 25 μg of protein was dried down in a SpeedVac^TM^ concentrator (Thermo Fisher Scientific, Waltham, MA, USA) and resuspended in 8 M urea in 100 mM ammonium bicarbonate. To these solutions, 5 mM TCEP and 50 mM iodoacetamide was added for 45 min at room temperature to reduce and alkylate cysteine residues. Proteins were then precipitated by 20% (*v*/*v*) trichloroacetic acid and allowed to sit at −20 °C for 1 h. Samples were then centrifuged at 15,000 rpm for 20 min at 4 °C, with the supernatant removed. Pellets were washed with cold methanol and centrifuged at 15,000 rpm for 20 min at 4 °C twice. Samples were dried in a SpeedVac^TM^ concentrator, resuspended in 10 μL 8 M urea in 100 mM ammonium bicarbonate, diluted to 1 M urea with 100 mM ammonium bicarbonate, and digested overnight at 37 °C, with the addition of 1:100 (*w/w*) ratio of trypsin. Peptides were desalted by a C18 Omix Tip (Agilent Technologies, Santa Clara, CA, USA) by a standard protocol and analyzed as described below.

### 2.3. Enrichment and Identification of S-Sulfurated Proteins in E. faecalis

This analysis was carried out essentially as previously described [7,30]. Wild-type *E. faecalis* was grown in 300 mL cultures as described above, and the experiment was performed in biological triplicate. At OD_600_ ≈ 0.2, cultures were collected by centrifugation for untreated cells or treated with 0.2 mM Na_2_S for 30 min followed by centrifugation. All cells were washed with ice-cold PBS three times and stored at −80 °C. Pellets were treated as previously described [7,30]. Samples were subjected to an LC MS/MS analysis as described below. Three biological replicates were used as treated (WT + Na_2_S) and untreated (WT) samples. Sigma ratios (σ^R^) were calculated for each protein using total area for cysteine peptides [30]. The fold change of *S*-sulfurated peptides (WT + Na_2_S/WT) were normalized to the protein abundance from data acquired in the proteomic analysis described above. *S*-sulfurated peptides only detected in one condition (WT or WT + Na_2_S) or proteins not identified in our no-enrichment proteomic analysis were assigned peaks areas equal to the lowest detected peptide or protein to eliminate division by zero during normalization [30].

### 2.4. LC-MS/MS Analysis of Proteome and S-Sulfurated Proteins

An LC-MS/MS analysis was carried out in the Laboratory of Biological Mass Spectrometry at Indiana University as previously described [30]. The resulting data were searched against the *E. faecalis TX4248* database (Uniprot UP000004846, 3273 entries) in Proteome Discoverer 2.1. Carbamidomethylation (CAM) of cysteine residues was set as a fixed modification. Protein N-terminal acetylation, oxidation of methionine, protein N-terminal methionine loss, protein N-terminal methionine loss and acetylation, and pyroglutamine formation were introduced as variable modifications. A total of three variable modifications were allowed. Trypsin digestion specificity with two missed cleavage was allowed. The mass tolerance for precursor and fragment ions was set to 10 ppm and 0.6 Da, respectively. The peptide peak area was quantified for MS1 ions and utilized to estimate fold-change in this label-free quantitation method.

### 2.5. Extraction and Measurement of Cellular Fatty Acids

Cell pellets from 14 mL cultures grown in biological triplicate as described above were resuspended in 1 mL methanolic NaOH [15% (*w/v*) NaOH in 50% (*v*/*v*) methanol] transferred to a glass, screw cap culture tube, vortexed for 5–10 s and saponified at 100 °C for 30 min. A total of 15 μg of pentadecanoic acid was added as an internal standard followed by fatty acid esterification, achieved by the addition of 2 mL of 3.25 N HCl in 46% (68%) (*v*/*v*) methanol at 80 °C for 10 min followed by extraction into 1.25 mL of 1:1 (*v*/*v*) methyl-*tert*-butyl ether in hexane for 10 min on a rotor (Barnstead Thermolyne Labquake Shaker, American Laboratory Trading, East Lyme, CT, USA). The aqueous phase was removed, and the organic phase was washed with 3 mL 1.2% (*w/v*) NaOH for 5 min on a rotor. The organic phase was promptly removed and stored until analysis. Samples were analyzed by GC-MS, by injecting 2 μL onto an SLB-5 MS 30 m × 0.25 mm × 0.25 um column. A sample run consisted of 35 °C (2 min) to 250 °C at 10 °C/min, then to 320 °C (3 min) at 20 °C/min. Fatty acids were identified based on the retention time of authentic standards. The MS was operated in single ion mode (SIM) using ions at 74 and 55 *m/z*. Chromatograms were extracted using these *m/z* and integrated, and peak areas were used to determine the relative abundance of each fatty acid. Fatty acids were designated in X:Y form, where X is the total number of carbon atoms, and Y is the number of double bonds, and each is presented as a fractional abundance of total fatty acids determined by the fraction of each fatty acid (peak area) to total fatty acids (sum of all peak areas).

### 2.6. Quantitation of Acetyl-CoA

Wild-type *E. faecalis* strain OG1RF was grown as described above in 100 mL cultures, and quantitation was performed using modifications of procedures described previously [31]. At an OD_600_ of 0.2, 2 mL of culture was centrifuged for 5 min, with the supernatant discarded and used to determine protein concentration. The remaining cell culture was quenched by the addition of a 3:2 (*v*/*v*) glycerol saline solution (−20 °C) to a ratio of 1:4 (*v*/*v*). Quenched cultures were placed at −20 °C for 5 min before centrifuging at 7100× *g* for 30 min at −11 °C. The supernatant was discarded, and the pellets were combined; washed with 2 mL of a 1:1 (*v*/*v*) glycerol saline solution (−20 °C); centrifuged for 20 min at −11 °C, with the supernatant discarded, and washed once more. Pellets were resuspended in 1 mL of 15% (*v*/*v*) methanol (cold) and transferred into a fresh 1.5 mL centrifuge tube and centrifuged at 10,000× *g* for 15 min at −11 °C, discarding the supernatant.

Cells were lysed by resuspending the pellet in 1.5 mL of 60% (*v*/*v*) cold methanol (−20 °C) and freeze-thawed in liquid nitrogen, followed by placement on ice. The process was repeated 5 times, and the cell debris was pelleted by centrifugation at a maximum speed for 15 min at −11 °C, and the supernatant was transferred to a new tube. The pellet was washed with 500 μL of 60% (*v*/*v*) methanol and centrifuged, and the supernatant was combined with that from the first centrifugation. The solutions were then dried down in a Speedvac concentrator and resuspended in 150 μL of LC-MS grade water and filtered through a 0.22 μm spin filter. A total of 10 μL was injected onto an Agilent AdvanceBio peptide C18 column (2.1 × 150 mm, 2.7 μm) equipped to an Aquity UPLC system coupled to a Waters G2S mass spectrometer. Acetyl-CoA was eluted using an acetonitrile-based gradient, slightly modified from that previously described (Solvent A: 0% (*v*/*v*) acetonitrile, 50 mM ammonium acetate; Solvent B: 100% (*v*/*v*) acetonitrile) at a flow rate of 250 μL/min [32]. A 15-min gradient was as follows: 0–1 min, 3% B; 1–7 min, 3–25% B; 7–8 min, 25–40% B; 8–11 min, 40% B; 11–15 min, 3% B. Quantitation was achieved via external calibration using the authentic acetyl-CoA standard (Sigma, St. Louis, MO, USA) and MassLynx software (version 4.2, Waters Corporation, Milford, MA, USA). The analysis was performed in biological triplicate.

### 2.7. Cloning and Purification of Recombinant Ef Pta

*Ef* Pta was cloned into the pHis parallel expression plasmid using the *Nco*1 and *Sal*I restriction sites, and the resulting plasmid was transformed into *E. coli* BL21 (DE3), cultured in LB medium at 37 °C until the OD_600_ reached 0.6–0.8, induced with 1 mM IPTG and expressed at 16–20 °C for 16 h. Cells were harvested by centrifugation and stored at −80 °C. The cell pellet was resuspended in a lysis buffer containing 25 mM Tris-HCl, 500 mM NaCl, 2 mM TCEP, 10% (*v*/*v*) glycerol and pH 8.0 and lysed by sonication. The cell lysate was clarified by centrifugation, and 70% (*w/v*) ammonium sulfate was added to salt out *Ef* Pta. The ammonium sulfate pellet was resuspended in the lysis buffer and was loaded on a HisTrap HP Ni-NTA column (Cytiva), which was pre-equilibrated with lysis buffer. The column was washed with lysis buffer followed by elution with an imidazole gradient from 0 to 500 mM. Fractions containing *Ef* Pta were combined, concentrated and further purified by size exclusion chromatography (Superdex G200 16/60 column, Cytiva) in a lysis buffer. Final fractions containing Pta with a purity of >95% estimated by SDS-PAGE were combined and stored with 10% (*v*/*v*) glycerol at −80 °C until use.

### 2.8. Enzyme Assays of Ef Pta Activity

*Ef* Pta is a reversible enzyme that can utilize either acetyl-CoA and inorganic phosphate or CoA and acetyl-phosphate as substrates, and both the forward and reverse reactions were assayed. *Ef* Pta was buffer-exchanged into fully degassed 25 mM Tris-HCl and 40 mM KCl at pH 8.0 in an anaerobic chamber. The acetyl-CoA-utilizing, CoA-forming reaction was measured using DTNB to derivatize the CoA thiol, and the TNB anion released was quantified by UV-Vis at 412 nm (ε = 14,150 M^−1^ cm^−1^), as described previously [33]. The 150 μL reactions contained 50 nM *Ef* Pta, 5 mM KH_2_PO_4_, acetyl-CoA ranging from 10 μM to 5 mM and 500 μM DTNB, buffered by 25 mM Tris-HCl and 40 mM KCl at pH 8.0, at room temperature in an anaerobic chamber. The reactions were initiated with the addition of the enzyme and incubated for 2 min, at which time the absorption at 412 nm was measured. To measure the reverse reaction, the 150 μL reactions contained 50 nM *Ef* Pta, 5 mM acetylphosphate and CoA ranging from 10 μM to 5 mM, buffered by 25 mM Tris-HCl and 40 mM KCl at pH 8.0 at room temperature in an anaerobic chamber. The reactions were initiated by the addition of the enzyme and incubated for 30 s, and the production of acetyl-CoA was monitored via formation of the thioester bond at 233 nm (ε = 4400 M^−1^ cm^−1^) [33,34].

To measure the production of acetyl-CoA, 50 μL reactions were prepared as described above. These reactions contained 20 or 150 μM of CoASSH from an in situ preparation [12] which contained Na_2_S, CoA disulfide, CoA and CoASSH. This was compared to reactions containing 20 or 150 μM CoA in the absence or presence of Na_2_S or CoA disulfide added at a concentration equal to that present in the in situ CoASSH preparation. The reactions were initiated with the addition of *Ef* Pta, incubated at room temperature for 30 s and quenched by addition of 150 μL 2 mM HPE-IAM in acetonitrile (1.5 mM final concentration). Samples were placed at −20 °C for 2 h to precipitate the protein and centrifuged at 4 °C for 20 min, and the supernatant was transferred to a clean 1.5 mL tube and dried down in a SpeedVac concentrator. Pellets were resuspended in 50 μL of HPLC grade water and then diluted at 1:20–1:100 in HPLC grade water and analyzed by LC-MS as described above. The concentration of acetyl-CoA was quantified by a series of authentic standards as described above.

### 2.9. Statistical Rationale and Bioinformatics Analysis

Proteins detected in fewer than two biological replicates were excluded from a statistical analysis of proteomic data, completed using an unpaired, two-tailed Student *t*-test with Welch’s correction. Functional information for the selected proteins were gathered from the National Center for Biotechnology Information (NCBI; https://www.ncbi.nlm.nih.gov/, accessed on 3 September 2019), BioCyc (https://biocyc.org/web-services.shtml, accessed on 3 September 2019) and Kyoto Encyclopedia of Genes and Genomes (KEGG; https://www.genome.jp/kegg/pathway.html, accessed on 3 September 2019) databases. Metabolism pathway information for the selected proteins was obtained from KEGG, and pathway analysis of *S*-sulfurated proteins was performed using the KEGG pathway mapper (https://www.genome.jp/kegg/mapper.html, accessed on 3 September 2019). A motif analysis of *S*-sulfuration sites was performed using pLogo (http://plogo.uconn.edu, accessed on 3 September 2019) [35]. For each modified cysteine, a 21-amino-acid sequence containing the *S*-sulfurated cysteine, with 10 amino acids flanking sequences on either side of the Cys, was selected. A background data set was constructed similarly using all cysteines identified in the reference genome for *E. faecalis* OG1RF (GenBank accession no. NZ_CP025020.1).

### 2.10. Cloning and Purification of Recombinant EfCoAPR

Gene expression and protein production were performed as previously described [12] with slight modifications. In brief, the *coaP* gene from *E. faecalis* strain OG1RF was fused with a hexahistidine tag and cloned into the pHIS parallel expression plasmid. The expression plasmid was transformed into *E. coli* BL21(DE3) and cultured in LB medium at 37 °C. *Ef*CoAPR expression was induced with 1 mM of IPTG for 16 h at 37 °C. After cell lysis by sonication, DNA removal, protein precipitation by ammonium sulfate and resuspension, His-tagged *Ef*CoAPR was purified by Ni-NTA affinity chromatography followed by size exclusion chromatography on a Superdex G200 HiLoad 16/600 (Cytiva, Marlborough, MA, USA) column using 20 mM MES at pH 6.0, 30 mM NaCl and 5% (*v*/*v*) glycerol buffer. Pure *Ef*CoAPR fractions were pooled, concentrated to ≈23 mg mL^−1^ using an Amicon Ultra 30 MWCO (Millipore, Burlington, MA, USA) by repeated concentration steps at 3500× *g* using an Eppendorf 5804R centrifuge and stored at −80 °C for further experiments.

### 2.11. X-ray Crystallography

Prior to crystallization experiments, *Ef*CoAPR was quickly thawed in a water bath at 42 °C and subjected to ultracentrifugation for 1 h at 217,200× *g* (Optima TL-100, TLA-100.3 fixed-angle rotor, Beckman Coulter). Initial crystallization screening was performed using a Mosquito LCP (SPT Labtech, Cambridge, UK) liquid dispenser robot and commercially available screens, e.g., JCSG +, BCS Screen, PACT Premier and Shot Gun, (Molecular Dimensions, Rotherham, UK) at protein concentrations ranging from 8–23 mg mL^−1^ using the vapor diffusion sitting-drop method. Poorly diffracting (8–10 Å) initial crystal hits from these screens were optimized using a cross-seeding approach as previously described [36] and the Seed Bead Steel Kit (Hampton Research, Aliso Viejo, CA, USA) following the manufacturer’s instructions. Crystals appeared in several conditions, and a scale-up experiment was performed using a gradient of 18–22% (*v/v*) PEG 3350, 0.1 M Bis-Tris propane pH 6.4–6.6 and 0.2 M NaI or NaF. Native crystals were grown under 20% (*v*/*v*) PEG 3350, 0.1 M Bis-Tris propane buffer at pH 6.50 and 0.2 M sodium iodide; cryo-protected and sent to the European Synchrotron Radiation Facility (ESRF, Grenoble, France) [37] for data collection. Diffraction data were indexed, integrated and scaled within the autoPROC data processing pipeline [38,39], and data quality was assessed with the phenix.xtriage tool within the PHENIX suite of programs [39,40], with the structure solved by molecular replacement (MR) with PHASER [41] as implemented in PHENIX, using one chain (monomer) of the X-ray structure of CoADR-RHD from *Bacillus anthracis* (PDB 3ICS) [42] as the search model, devoid of any cofactors, solvent molecules and other ligands. Iterative model building and refinement were carried out in a cyclic manner with phenix.refine [43], BUSTER-TNT and COOT [44], until a complete model was built and refinement convergence achieved. The *Ef*CoAPR model was validated with MolProbity [45] within PHENIX with the atomic coordinates and structure factors deposited in the Protein Data Bank (http://wwpdb.org/, accessed on 3 September 2019) under accession code 8A56. Structural illustrations were rendered using PyMOL (Version 2.4.1, Schrödinger, LLC., New York, NY, USA) and COOT [43].

### 2.12. Molecular Dynamics (MD) Simulations

Simulations were carried out on *Ef*CoAPR with both cysteine residues unmodified and bound coenzyme A in its reduced state (CoASH) using GROMACS 2020.3 [46]. To build this system, we used atomic coordinates from the structure above (PDB 8A56) with all crystallographic water molecules in place and FAD coordinates as defined in the structure. Since only the PAP moiety of CoA could be resolved in the experimental electron density, we placed CoASH into the structure by superimposition with CoADR-RHD from *Bacillus anthracis* (PDB 3ICS) [42]. The atomic coordinates for the E474-Q479 loop from the B subunit were placed and refined using MODELLER [47]. The force field parameters were developed on the basis of the CHARMM General Force Field (CGenFF) [48]. The atom types and initial parameters were determined using the CGenFF webserver https://cgenff.paramchem.org, accessed on 3 September 2019) [48,49]. Parameters for FAD and CoASH were obtained from an electronic structure calculation using Gaussian03 at the HF/6–31G* level basis set, followed by a derivation of partial atomic charges using the RESP procedure [50]. The complex was placed into a truncated octahedral box of TIP3P water molecules, defining a distance of 15 Å between the border of the box and the closest atom of the solute. This gave us a total of 157,813 atoms in the system, with 46,933 water molecules and 1089 protein residues. The system was neutralized with Na^+^ ions, and Na^+^ and Cl^−^ ions were added to 0.15 M ionic strength.

Geometric optimization was performed with an energy minimization step of 5000 cycles with a 1000 kJ mol^−1^ nm^−1^ force constant applied over all atoms, excluding water molecules. Afterwards, the temperature was increased from 0 to 10 K using a Berendsen thermostat [51] with a coupling constant of 0.05 ps, in a 10 ps constant volume MD with a 0.1 fs time step, and a harmonic restraint potential of 1000 kJ mol^−1^ nm^−1^ applied over all protein residues and ligands. Thereafter, the temperature was increased from 10 to 300 K using a Berendsen thermostat (coupling constant of 0.05 ps), in a 200 ps constant volume MD with a 0.5 fs time step, applying a force constant of 1000 kJ mol^−1^ nm^−1^ to the protein backbone atoms and ligands. After the samples had been heated, the density was equilibrated with a 200 ps MD simulation at constant temperature (300 K) using V-rescale as a thermostat [52] and pressure with a time step of 1 ps (NTP ensemble), while applying a force constant of 1000 kJ mol^−1^ nm^−1^ to heavy atoms on the protomers and ligands. Additionally, another 10 ns MD of NTP ensemble was performed, with positional restraints on Cα atoms and the ligands. For production MD, 1400 ns simulations in the NTP ensemble were conducted, with a time step of 2 fs, under periodic boundary conditions [53] using the LINCS algorithm [54] to constrain all bonds. Long-range electrostatic interactions were handled with PME [55], setting a cutoff distance of 12 Å. All molecular visualizations and drawings were performed with the Visual Molecular Dynamics program [56].

## 3. Results

### 3.1. Global Proteomics Profiling of Wild-Type E. faecalis before and after Addition of Exogenous Na_2_S

To elucidate the global response of the proteome to exogenous sulfide, we employed a label-free “bottom-up” proteomics analysis on soluble lysates to estimate the change in cellular abundance of specific proteins (Appendix A). In total, 1115 proteins were detected at least twice in four replicates, with 50 proteins observed only in the wild-type (WT) + Na_2_S cells and 24 observed only in WT untreated cells (Figure 1A). Fifty-two proteins (4.9%) detectable in both WT and WT + Na_2_S cells exhibited a significant change in cellular abundance, defined here as >2-fold change and *p* < 0.001, which includes the *cst*-like operon encoded enzyme CoAPR (Figure 1B). Strikingly, CoAPR is detected in untreated cells and is among the 25% least-abundant proteins that are detected; this suggests that the RSS-sensing transcriptional repressor of the *cst*-like operon CstR “senses” endogenous RSS prior to further induction by the exogenous sulfide (Figure 1C). The presence of CoAPR in unstressed WT cells is also consistent with the low endogenous level of coenzyme A persulfide (CoASSH) in this organism, as CoAPR specifically reduces CoASSH to the free thiol and H_2_S [12]. After addition of Na_2_S, CoAPR rises to the top 20% most abundant proteins detected consistent with the regulatory model of CstR (Figure 1D) [12,27].

In addition to CoAPR, two transcriptional regulators, Rex (OG1RF_12010) and VicR (OG1RF_10965); several uncharacterized proteins and proteins involved in translation, including nine tRNA synthetases, were also increased in the proteome after addition of exogenous Na_2_S (Figure 1B). Rex is NADH-regulated, contributes to NAD ^+^/NADH homeostasis and impacts H_2_O_2_ detoxification in *E. faecalis* [57], while VicR is the soluble response regulator of the VicRK two-component membrane sensor kinase, the kinase activity of which is modulated by extracellular glutathione and dithiothreitol, a model reductant [58]. The concomitant increase in catalase (OG1RF_11314) and decrease the cellular abundance of FeoA, a known Rex target [57] (Appendix A), collectively suggests an overall change in the cellular redox potential with exogenous sulfide. Of additional interest is the increase in the cell abundance of dephospho-CoA kinase (DPCK) and two thioesterases (OG1RF_11875, OG1RF_11277) of unknown substrate specificity that catalyzed the hydrolysis of a thioester bond. DPCK catalyzed the final step in the de novo biosynthesis of CoA, which might suggest increased production of CoA under these conditions; this finding is consistent with the ≈1.5-fold increase in CoA 30 min post-addition of exogenous sulfide found in previous work [12]. In addition, an uncharacterized multidomain sulfurtransferase (STR, OG1RF_10483), lipoate-protein ligase (LplA, OG1RF_12105) and a candidate cysteine desulfurase (IscS, OG1RF_10258) involved in Fe-S cluster biogenesis were also significantly increased after addition of exogenous Na_2_S. Two of these enzymes are known or projected to be involved in persulfide transfer (STR, IscS), while LplA lipoates enzymes that couple acyl-transfer from acetyl-CoA in large multienzyme dehydrogenases [59].

### 3.2. Proteome Persulfidation in E. faecalis

To explore H_2_S/RSS signaling in *E. faecalis* further, we utilized our previously developed enrichment-reduction strategy [7,30] to profile proteome persulfidation in wild-type *E. faecalis* in the absence or presence of exogenous Na_2_S. The extent to which a Cys residue is persulfidated after the addition of exogenous Na_2_S vs. prior is reflected in the parameter σ^R^. A σ^R^ value of 1.0 corresponds to peptides that are not persulfidated prior to the addition of Na_2_S (24 proteins), and a σ^R^ value approaching 0 indicates peptides that are only persulfidated in the absence of exogenous sulfide (13 proteins). We identified 356 proteins (≈13% of the proteome) as persulfidated, and the majority of these proteins (90%) were persulfidated in both untreated and Na_2_S-treated cells (Appendix A, Appendix A). This is consistent with previous studies of *S. aureus* [7] and *A. baumannii* [13] and suggests that the proteome acts as a reservoir of bioactive sulfane sulfur. These proteins map to many cellular processes, including fatty acid biosynthesis (FAS), amino acid biosynthesis, pyrimidine metabolism and central carbon metabolism. Efforts to identify a consensus sequence associated with sites of protein persulfidation in *E. faecalis* generally failed, even when we restricted our analysis to peptides with high σ^R^ (Appendix A).

A consistent challenge in profiling proteome persulfidation, as well as many other cysteine thiol redox modifications [60], is the ability to distinguish persulfidated cysteines that are regulatory from those that are persulfidated collaterally on solvent-accessible or highly reactive cysteines. Since σ^R^ analysis alone does not consider changes in protein abundance, we measured the change in protein abundance using a label-free method (Figure 1, Appendix A) and used this change to normalize the change in persulfidation. We found that the vast majority of proteins with σ^R^ values greater than or less than one standard deviation from the mean (σ^R^ = 0.52) were found in the “wings” of the normalized abundance plot, a finding consistent with an identical analysis in *A. baumannii* (Appendix A) [13]. A better way to express these data is to compare the change in the persulfidation status of a particular peptide to its change in abundance in a log–log plot (Figure 2) [30]. Here, we predicted that a persulfidated peptide with a significant increase in persulfidation status coupled with relatively small change in protein abundance may be strong candidate regulatory protein targets (Figure 2).

An evaluation of protein thiols for which we have total persulfidation and abundance data, i.e., detectable and persulfidated in both conditions, revealed that a majority did not have a statistically significant change in persulfidation status or fractional abundance. This supports the idea that the proteome functions as a sink for sulfane sulfur that is not greatly impacted by the addition of a high dose of exogenous Na_2_S to cells [7,13]. Those proteins that are more likely to represent regulatory targets via persulfidation are those that are only identified as persulfidated under conditions of Na_2_S treatment, the majority of which may exhibit only small statistically significant changes in cell abundance (Figure 2B). A striking exception to this was CoAPR, which exhibited a large increase in protein abundance after the addition of Na_2_S due to the transcriptional derepression by CstR [12] with only the cysteine derived from the C-terminal rhodanese domain (C508) identified as persulfidated in Na_2_S-treated cells (Appendix A).

Other persulfidated proteins of note in this analysis included phosphotransacetylase (Pta), which catalyzed the reversible interconversion of acetyl-CoA and acetyl phosphate, and cysteines in enzymes involved in type II fatty acid biosynthesis (FAS), including AccA, AccC and FabN (Figure 2B; Appendix A) [61,62]. AccA and AccC are part of a four-enzyme acetyl-CoA carboxylase (ACCase) complex, which defines the initial and committed step of FAS, carboxylating acetyl-CoA to make malonyl-CoA prior to linkage to the acyl-carrier protein (ACP) [62,63]. FabN is a bifunctional enzyme, possessing both 3-hydroxydecanoyl-ACP dehydratase activity, to create *trans*-2-decanoyl-ACP, and isomerase activity, converting *trans*-2-decenoyl-ACP to *cis*-3-decanoyl-ACP [64]. FabN is functionally analogous to, although structurally distinct from, FabA in *E. coli*, which defines a major point of regulation of the ratio of unsaturated to saturated fatty acids [61]. As such, *Ef* mutants lacking FabN exhibited unsaturated fatty acid auxotrophy, which could be partly rescued by supplementation by oleic acid (18:1) [65], revealing that *Ef* FabN is critically important in the biosynthesis of longer chain, unsaturated fatty acids [64]. Additionally, two other enzymes involved in FAS, including AccB (C119) of the ACCase complex and the primary β-ketoacyl-ACP synthase II, FabF (C262), were also persulfidated (Appendix A, Appendix A). If persulfidation is regulatory for any of these enzymes, sulfide stress may impact the cellular concentrations of acyl-CoA species or the cellular composition of membrane-derived fatty acids.

### 3.3. Exogenous Na_2_S Impacts Cellular Composition of Fatty Acids

To investigate the impact of exogenous sulfide on fatty acids and fatty acid biosynthesis, we extracted fatty acids from bacterial cultures before and after addition of Na_2_S and quantified them as methyl esters by GC-MS [66]. This analysis revealed that 14:0, 16:0, 18:0 and 18:1 fatty acids dominated the profile, with a statistically significant increase in the relative abundance of 14:0 and 16:0 fatty acids following the addition of Na_2_S (Figure 3A). We noted a corresponding decrease in the major unsaturated fatty acid species, 18:1 (*cis* and *trans* isomers were not resolved here), with little change in 16:1 species, and an accompanying decrease in the relative abundance of shorter chain saturated fatty acids (≤12:0), the latter of which was anticipated as a result of their elongation to longer chain saturated fatty acids (Figure 3A,B). In addition, the ratio of total saturated to unsaturated fatty acids increased over time in cells treated with exogenous sulfide relative to untreated WT cells (Figure 3C). When we compared the relative abundance of each fatty acid between WT and WT + Na_2_S, we found that shorter chain fatty acids up to 14:0 followed the same trend, while longer fatty acids (>14:0) showed significant differences between the two conditions (Appendix A). Together, these changes contribute to the observed difference in the ratio of total saturated to unsaturated fatty acids.

### 3.4. Exogenous Na_2_S Impacts Acetyl-CoA via Enzyme Inhibition of Phosphotransacetylase (Pta) by CoASSH

Our fatty acid profiling results revealed a significant impact on the relative abundance of saturated vs. unsaturated fatty acids, consistent with regulatory inhibition of FabN activity (Figure 3 and Appendix A). Persulfidation of three of the four subunits of ACCase motivated us to investigate a potential impact on acyl-CoA species under these conditions, since in *E. faecalis,* acetyl-CoA and malonyl-CoA are the predominant acyl-CoA species that are required for FAS (Appendix A). Malonyl-CoA was found to be below our level of detection in both untreated and Na_2_S-treated cells. In striking contrast, acetyl-CoA exhibited a nearly 20-fold increase 60 min post-addition of exogenous Na_2_S (Figure 4A) in wild-type cells. This accumulation of acetyl-CoA was ≈12-fold higher than that quantitated from control WT cells grown for 60 min without addition of Na_2_S (Figure 4A). This increase in acetyl-CoA may have originated from increased activity of acetyl-CoA producing enzymes, e.g., pyruvate dehydrogenase (PDH), acetate kinase (AckA) and Pta (Figure 4B), or decreased activity of the acetyl-CoA carboxylase (ACCase) complex [61,62].

We chose *E. faecalis* Pta to investigate the extent to which the observed accumulation of acetyl-CoA could be traced to the combined activities of AckA and Pta, since Pta was identified as extensively persulfidated only under conditions of exogenous Na_2_S treatment (Figure 2). The site of persulfidation in this enzyme, C182, is located on a loop in the CoA/acetyl-CoA binding pocket, as defined by a *Methanosarcina thermophila* Pta-CoA co-crystal structure [69]. Persulfidation at this location could have a regulatory impact on enzyme activity (Figure 4C). We purified *E. faecalis* Pta and characterized both the forward (acetyl-phosphate-forming) and reverse (acetyl-CoA-forming) reactions, which collectively showed that recombinant *Ef*Pta is a functional enzyme, with kinetic parameters comparable to those reported for other Pta enzymes in the literature (Appendix A) [33,70,71].

We next investigated the extent to which *in situ*-prepared CoASSH could function as an inhibitor of Pta activity. The yield of acetyl-CoA was measured by HPLC from reactions containing Pta; acetyl-phosphate and either its natural substrate, CoASH, or a CoASSH-containing mixture [12], which included equimolar CoASH and excess Na_2_S and CoA disulfide. Control reactions were also performed, which included CoASH and either Na_2_S or CoA disulfide at concentrations equivalent to those present in the in situ CoASSH mixture. At low concentrations of substrate (CoASH; 20 µM), the yield of acetyl-CoA was similar among reactions containing only CoA and CoASSH and in control reactions (Figure 4D). At high concentrations of the substrate (150 µM, above the *K*_m_; see Appendix A), however, there was little detectable production of acetyl-CoA in the presence of the CoASSH mixture. This inhibition of activity was due to CoASSH specifically, since there was no negative impact on product formation from the control reactions containing excess Na_2_S and CoA disulfide (Figure 4D). We did not investigate if Pta was persulfidated by CoASSH under these conditions, but it proved difficult to persulfidate Pta with various other sulfur donors, e.g., inorganic polysulfide, in vitro using standard methods (data not shown) [7]. These findings with Pta are consistent with a CoASSH “poisoning” model proposed previously (Figure 4E), where CoASSH functions as substrate mimic that forms long-lived complexes with CoA- and short-chain acyl-CoA-requiring enzymes [29,72]. If a nearby thiol is present, this may result in persulfide transfer and subsequent inhibition of enzyme activity.

### 3.5. Crystallographic Structure of Ligand-Bound EfCoAPR

In order to better understand the molecular basis of cellular resistance to CoASSH poisoning described above and likely mediated by CoAPR in *E. faecalis* (Figure 5A) [12], we determined the crystallographic structure of the enzyme as isolated from *E. coli* to a 2.05 Å resolution by molecular replacement using *Ba*CoADR-RHD [42] as the search model (Figure 5B; see Appendix A for structure statistics). The asymmetric unit is composed of two molecules of *Ef*CoAPR in a tightly packed dimeric arrangement, consistent with a homodimer assembly state [42] and containing two FAD molecules and two molecules of 3′-phosphate-adenosine-5′-diphosphate (PAP) (Figure 5C), with the latter found in a region where CoA is expected to bind [42] (Appendix A). While inspection of the electron density revealed that the PAP moiety of CoA was well-ordered, the remainder of the CoA molecule could not be placed in our structure, which suggests considerable flexibility in the pantothenate portion of the ligand [42] (see below). The modelled PAP moiety was ~13.3 Å away from FAD riboflavin moiety, which was fully modelled and refined.

Each subunit is composed of two functional domains (Figure 5B). The N-terminal CoA disulfide reductase (CDR) domain consists of residues 1–446 and harbors the first catalytically active thiol, C42, while the C-terminal sulfurtransferase (rhodanese) domain is composed of residues 447–544 and contains the second catalytically required cysteine, C508 [12], which is persulfidated in cells (Figure 2). The dimer interface is extensive (3600 Å^2^) and is largely stabilized by hydrophilic residues, contributed primarily by the CDR domains, with the FAD bound such that its ATP portion is close to the “bottom” of each protomer, with the isoalloxazine ring in van der Waals contact with the S^γ^ of C42. The two sulfurtransferase domains of CoAPR are on the periphery of the dimer, effectively wedged between the two reductase domains. The 3′-phosphoadenosyl moiety of the PAP portion of the bound CoA forms an extensive part of this interface, deeply buried in the sulfurtransferase domain 10.7 Å from C508, wedged between the α1 helix of the opposite protomer, the terminal helix of the reductase domain of the same protomer and the core helix of the rhodanese domain (residues 512–525), which is immediately C-terminal to C508 (Figure 5B). C42 and C508′ make their closest approach from opposite protomers, but they are 26.6 Å apart.

The *Ef*CoAPR structure is similar to other FAD-dependent pyridine nucleotide-disulfide oxidoreductases, including those classified as CDRs, NADH (per)oxidases, NAD-dependent persulfide reductases (Npsr) [73,74] and *Bacillus anthracis* CoADR-RHD (*Ba*CoADR-RHD, 3ICS) [42]. In striking contrast to *Ef*CoAPR, the entire CoA molecule is visible in these three structures, with the CoA sulfur atom in close proximity to the flavin isoalloxazine ring of the opposite protomer [42,74,75]. The high structural similarity between *Ef*CoAPR and *Ba*CoADR-RHD models revealed nearly perfect superposition of the PAP moiety of CoA and FAD cofactors (Appendix A). The main structural differences between these models are found in two loop regions, in particular, the E474-Q479 loop in the rhodanese domain, which is disordered in *Ef*CoAPR but α-helical in *Ba*CDR-RHD (Figure 5B). This 474-Q479 loop appears to form a physical barrier that may gate direct access of C508 to the active site cavity. A tunnel ≈25 Å in length connects the aperture of the solvent-exposed catalytic C508 and the CoA cavity located in the rhodanese domain to the C42 and FAD binding groove, located in the CDR domain of the opposite protomer (Figure 5D and Appendix A). This tunnel may well be an essential feature of turnover, as it connects the FAD and CoA binding pockets to the solvent (see below).

### 3.6. Molecular Dynamics Simulations

To obtain additional insights into the dynamics of this complex [42], we carried out molecular dynamics simulations of our FAD- and CoASH-bound structure after building the remainder of CoASH into this model (Figure 6A). Two replica simulations of 1.4 µs each were performed, and in both runs, the pantothenate moiety of the CoA molecule corresponding to the same binding site (protomer B) left the vicinity of C42 (Figure 6A) and moved towards C508′ in the rhodanese domain of the other protomer (Figure 6B; Appendix A). In striking contrast, the pantothenate arm of the CoA bound to the adjacent protomer (protomer A) remained close to C42, routinely reaching distances below ≈3.5 Å (Appendix A). This suggests that each subunit might stabilize the CoA molecule in either an “extended” (Figure 6A) or “bent” (Figure 6B) conformation, a finding that might imply an “alternating active sites” model of catalysis, in which one subunit turns over while the other rests. The per-residue root mean-square fluctuation (RMSF) calculated over these simulations showed that the rhodanese domain presents distinct fluctuations between protomers (Appendix A), consistent with a concerted movement that may link CoASSH formation, folding of the E474-Q479 loop and significant dynamics associated with the rhodanese region (Appendix A). The relative position PAP moiety from CoA is stable along the MD trajectory, and only the pantothenate arm changes its position (Figure 6A,B and Appendix A); furthermore, we found that the “arm” makes two slightly different approaches to C508 on opposite sides of the R513 side chain, with a minimum S-S distance of ≈6.5 Å (Figure 6C,D). The persulfidation of C508 may then bring these two reactive groups in sufficiently close proximity to perform the persulfide transfer.

These MD simulations provide significant support for the “swinging pantothenate arm” hypothesis [57], where the tightly bound CoA attacks a persulfide on C508, bringing CoASSH back to the reductase active site; the CoASSH is then attacked by the C42 with the release of the H_2_S and formation of mixed disulfide, which is subsequently reduced by the flavin, which itself is reduced by a more weakly bound NAD(P)H [74]. Significant levels of C508 persulfidation in cells on par with those of candidate persulfidation targets Pta and FabN (Figure 2), coupled with no observable C42 persulfidation, are fully consistent with this mechanistic model. Whether or not the CoASH product is released was unclear from our structure and simulations. Given the extensive contacts in what might be a “closed” conformation (Figure 5B,D), the release of CoASH would seem to require access to a transiently “open” form that would disrupt the extensive interdomain interface into which PAP is wedged (Figure 5D). If CoASH is not released with each turnover, this would require high specificity in the initial persulfide transfer from CoASSH to the C-terminal sulfurtransferase domain; indeed, this domain possesses weak thiosulfate sulfurtransferase activity, suggesting that some other thiol persulfide, e.g., CoASSH, might be a better substrate for this persulfide transferase activity [12].

## 4. Discussion

The gastrointestinal tract is host to hundreds of microorganisms that have developed a symbiotic relationship with host epithelial cells in this niche based, in part, on the reduction of sulfur-containing molecules, e.g., sulfate, thiosulfate and tetrathionate, and the oxidation of H_2_S. Perturbation of this symbiotic relationship results in gastrointestinal disease [20,21,22] due to increased H_2_S or other sulfur species, e.g., tetrathionate, which provide a growth advantage to some pathogens in this niche [76]. *E. faecalis* encodes a CydAB, the alternate oxidase that is far less susceptible to H_2_S-mediated inhibition [77,78,79], which may allow this organism to respire under normal and infection-relevant H_2_S concentrations in the gut; indeed, the cell abundance of CydD, an assembly factor for CydAB biogenesis, increases under these conditions (Appendix A). In addition, *E. faecalis* CstR, which regulates the *cst*-like operon, may provide a growth advantage in this niche by lowering ambient persulfidation [28], but this is currently unknown. Here, we employed exogenous sulfide to investigate the proteomic response of *E. faecalis* to sulfide mis-regulation and profile protein persulfidation to determine potential targets of H_2_S/RSS modification in this organism.

Our proteomic analysis revealed that ≈5% of the proteome that was detected under our conditions exhibited a significant change in normalized fractional abundance in response to exogenous sulfide. In fact, the *cst*-like operon-encoded protein CoAPR [12] was detectable in a soluble lysate in WT untreated cells, which may explain the significantly low endogenous concentrations of CoASSH compared to other low-molecular-weight persulfides (Figure 1). CoAPR then became one of the top 20% most abundant proteins detectable post-addition of Na_2_S. This strongly suggests that *E. faecalis* protects the integrity of the CoA pool by preventing over-persulfidation of this important metabolite, with CoAPR being a significant component of this adaptive response. Consistent with this idea, we observed a significant increase in the cell abundance of DPCK, which is required for the biosynthesis of CoA, and two uncharacterized thioesterases, which may be deployed to hydrolyze an acyl-CoA substrate(s) to release CoA to meet cellular needs under these conditions (Figure 1B).

We found that ~13% of the proteome in *E. faecalis* harbored persulfidated cysteines, and the extent to which persulfidation changed relative to protein abundance was significant for only a handful of proteins (Figure 2). Proteins for which persulfidation status changed significantly relative to changes in protein abundance included CoAPR and its transcriptional regulator, CstR, as anticipated [12]. CstR is a di-thiol-containing transcriptional repressor, and our persulfidation profiling revealed that both cysteines, C31 and C62, in *E. faecalis* CstR were persulfidated in the cell, consistent with the capture of either a reaction intermediate or the product of the reaction of H_2_S on a trisulfide bridge (C31-S-S-S-C62′), resulting in a persulfide adduct on each Cys residue [28]. Of the two cysteines in CoAPR, only the C-terminal rhodanese domain cysteine (C508) was persulfidated in vivo, consistent with a role in persulfide transfer in a “swinging pantothenate arm” reaction mechanism proposed by others [42,75] and for which we provided significant support from MD simulations (Figure 6). Indeed, the two active sites in CoAPR were separated by ≈27 Å across protomers (Figure 5B), and both are required by turnover and full product formation [12]; since the two active sites are spanned end-to-end by the CoA molecule, this solves the problem of getting these two active sites close enough to perform direct persulfide shuttling to create the anticipated mixed disulfide with C42 close to the flavin [12] (Figure 5 and Figure 6). Additional work will be required to understand the specificity of persulfidation of C508, CoA product release (if any), the metabolic fate of the so-generated H_2_S and the extent to which the rhodanese domains pack against the reductase dimer core in the absence of CoA, i.e., adopts an “open” conformation.

Several enzymes involved in the biosynthesis of fatty acids and acyl-CoA species exhibit significant changes in the abundance-normalized persulfidation status, including AccA, AccC, FabN and Pta (Figure 2). Persulfidation of these proteins may function as a regulatory switch via modification of a cysteine thiol, which impacts enzyme structure or turnover. Indeed, fatty acid profiling revealed significant changes in the relative abundance of fatty acids in response to exogenous sulfide, both in length and degree of saturation (Figure 3 and Appendix A). These differences were largely derived from a significant increase in the most abundant fatty acid, 16:0, and a decrease in one longer (18:0) and three unsaturated (16:1*_cis_*, 16:1*_trans_* and 18:1) fatty acids. Since the primary role of fatty acids is to form the phospholipid bilayer, these changes could induce significant alterations in the composition and physical properties of the cell membrane [80,81,82]. These changes in lipid composition may well be due to a negative regulatory effect of persulfidation of C21 in FabN, the deletion of which is known to impact the biosynthesis of unsaturated fatty acids [65]. Although there is no structure of *Ef* FabN, an AlphaFold2 model [67,68] revealed a “hot-dog” fold reminiscent of FabZ hexamers from *H. pylori* [83] and *P. aeruginosa* [84] and revealed that C21 is located in the α1-β loop, which is highly conserved in FabZs, close to the hot-dog (α3) helix, near the active site glutamate residue and the trimer interface, in this trimer of dimers architecture (Appendix A) [85]. Small displacements of the the α3 helix are known to impact enzyme specificity [85]; alternatively, persulfidation of C21 may lead to disassembly of the FabN hexamer to dimers and/or influence the way in which holo-ACP engages the hexamer [86].

The high levels of persulfidation of C50 in the biotin carboxylase (BC; AccC) subunit, which is conserved in the *E. coli* enzyme, may also be functionally important, since C50 forms a hydrogen bond at the interface with the biotin carboxylase carrier protein (BCCP; AccB) subunit of the heterooctameric *E. coli* BC_4_•BCCP_4_ biotin carboxylase complex [87,88]. Persulfidation, here, may destabilize the quaternary structure of the BC complex, thus inhibiting ACCase activity at this committed step of FAS. In addition, the AccA and AccD subunits form a heterotetrameric α_2_β_2_ carboxyltransferase enzyme [89], and C114, which is highly persulfidated in H_2_S-treated cells, is located in the acetyl-CoA binding cleft, very close to the thiol/thioester end of the bound substrate (Appendix A). The extent to which these modifications are regulatory in vitro is not yet known, but the accumulation of cellular acetyl-CoA (Figure 4) and high cellular abundance of downstream FabH, which supplies acetoacetyl-CoA in the initiation of FAS (Appendix A), may be reporting on reduced flux at early stages of FAS in Na_2_S-stressed cells.

Membrane composition is known to impact antimicrobial activity [90] in bacteria, and in *E. faecalis,* the incorporation of exogenous fatty acids has been shown to provide protection from membrane damaging agents, including antibiotics [66,91]. The loss of FabN specifically leads to a relative increase in saturated fatty acids incorporated into phospholipids [65]. This is known to decrease membrane fluidity and results in an increase in the resistance to antibiotics including daptomycin [92]. In addition, FabN mutants also appear to elicit a decreased inflammatory response during infections, and these features may be broadly immunoprotective [65]. Protein lipidation, primarily palmitoylation, requires palmitic acid (16:0). Although its role in bacteria is not fully understood, some studies have suggested that bacterial palmitoylation is associated with infections, activates toxins and “hijacks” host enzymes to modify their proteins [93,94,95,96]. These data suggest that *E. faecalis* may be capable of leveraging infection-relevant H_2_S/RSS to remodel the membrane in a way that is broadly protective against host stressors.

Finally, our finding of a significant accumulation of the major acyl-CoA species in *E. faecalis*, acetyl-CoA (Figure 4A), beyond FAS, may suggest that other enzymes that utilize or produce acetyl-CoA may well be subject to inhibition by CoASSH. Here, we demonstrated that CoASSH inhibits the acetyl-CoA-forming reaction by *E. faecalis* Pta, which suggests that the reverse reaction (acetyl-CoA utilizing) would also be inhibited (Figure 4D). These data with Pta, coupled with an analysis of AccAD carboxytransferase, are consistent with the hypothesis that CoASSH effectively poisons CoA- or acyl-CoA-requiring enzymes by binding as a substrate or product mimic, like that previously reported for human butyryl-CoA dehydrogenase [29]. CoASSH poisoning may result in persulfidation of a thiol in or near the active site, supported by the identification of several CoA and acyl-CoA requiring enzymes as extensively persulfidated in cells (Figure 4E), including Pta and AccAD (Appendix A). This model requires that a cysteine residue be proximate to the persulfide “end” of a bound CoASSH and participate in sulfur transfer from CoASSH to generate CoA and a persulfidated enzyme. Although more support for this model is required, we suggest that an important metabolic impact in *E. faecalis* in response to exogenous sulfide is that of over-persulfidation of the CoA pool. In *E. faecalis*, CstR-regulated CoAPR (Figure 5) likely functions to maintain the integrity of the CoA pool, while in *S. aureus*, CstR-regulated enzymes collaborate in some way to minimize endogenous CoASSH to a level less than 1–2% of the total pool, depending on sulfur source. While, at first glance, this may seem like a modest impact (in *E. faecalis*, ambient levels of CoASSH are ≤10% that of *S. aureus*, at ≤0.1%) [12], if much of the CoA is bound to enzymes in cells, this would place a highly reactive hydropersulfide group within or in close physical proximity to the active sites of CoA- and short-chain acyl-CoA-requiring enzymes. The extent to which CoASSH poisoning in *E. faecalis* and other bacteria is operative remains to be determined.

## 5. Conclusions

In this work, we leveraged a global map of proteome persulfidation in *E. faecalis* to identify and discuss strong candidate regulatory persulfidation sites in enzymes associated with acyl-transfer reactions and fatty acid biosynthesis. H_2_S/RSS mis-regulation induced significant perturbations in the fatty acid composition of the cell membrane and an increase of cellular acetyl-CoA, both consistent with a CoASSH poisoning model of primary acyl-CoA-dependent processes. The enzyme responsible for clearing toxic CoASSH in *E. faecalis* is CoA persulfide reductase (CoAPR), the structure and molecular dynamics analysis of which revealed a catalytic mechanism that is consistent with its persulfidation status in cells.

## Figures and Tables

**Figure 1 antioxidants-11-01607-f001:**
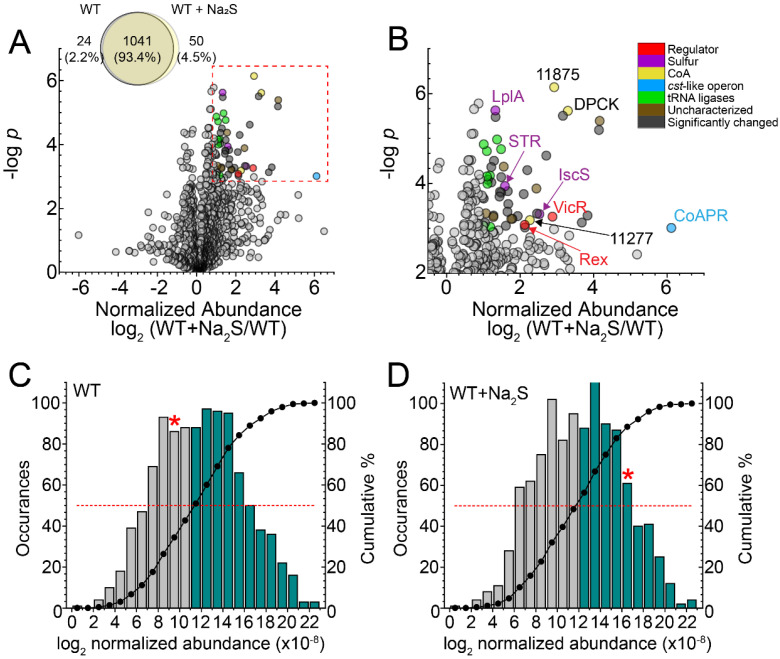
LC-MS/MS proteomics analysis of wild-type *E. faecalis*. Protein profiles of untreated (WT) vs. 0.2 mM Na_2_S-treated (WT + Na_2_S) *E. faecalis* cells from four biological replicates. (**A**) Venn diagram of proteins detected at least twice in four biological replicates, with volcano plot for proteins detected at least two times in four biological replicates. The significance threshold was set at *p* < 0.001 and fold-change in protein abundance at >2, as enclosed in the red dashed-line box. Circles corresponding to proteins within this significance threshold are shaded according to the annotated function in panel B. (**B**) Expanded view of proteins within a significance threshold, shaded according to the annotated function, with select proteins annotated with the protein name or locus tag (OG1RF_xxxxx). Histogram plot of the distribution of normalized abundance for all proteins detected in the (**C**) WT and (**D**) WT+ 0.2 mM Na_2_S strains. The top-most 50% abundant proteins are indicated with a red dashed line, with the bars shaded blue. The red star (*) indicates the cellular abundance of CoAPR.

**Figure 2 antioxidants-11-01607-f002:**
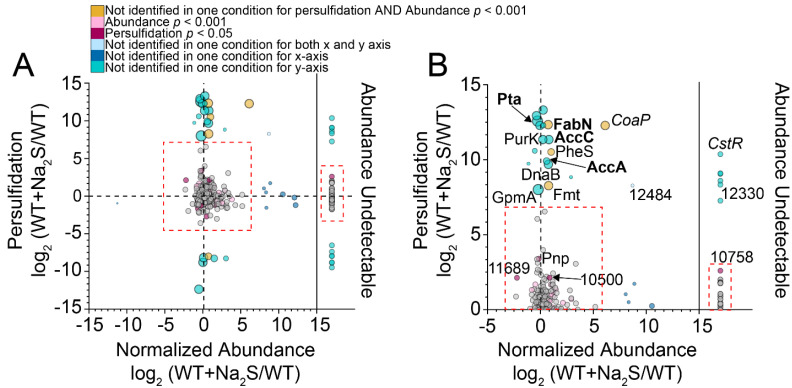
Changes in protein persulfidation relative to changes in protein abundance in *E. faecalis* before and after the addition of exogenous sulfide. (**A**) Log–log plot of the change in protein persulfidation (WT + Na_2_S/WT) vs. change in protein abundance (WT + Na_2_S/WT). Each circle represents a single protein identified as persulfidated, colored according to the significance threshold in the legend. The red dashed boxes include proteins whose persulfidation status and/or normalized abundance was detectable in both WT and WT + Na_2_S samples. Proteins outside these boxes were not detectable in one or more conditions and utilized a peak area one order of magnitude lower than the lowest detectable protein to calculate persulfidation or abundance changes. (**B**) Expanded view of proteins of which their persulfidation status increased after the addition of exogenous sulfide, with select proteins annotated with the protein name or locus tag (OG1RF_xxxxx).

**Figure 3 antioxidants-11-01607-f003:**
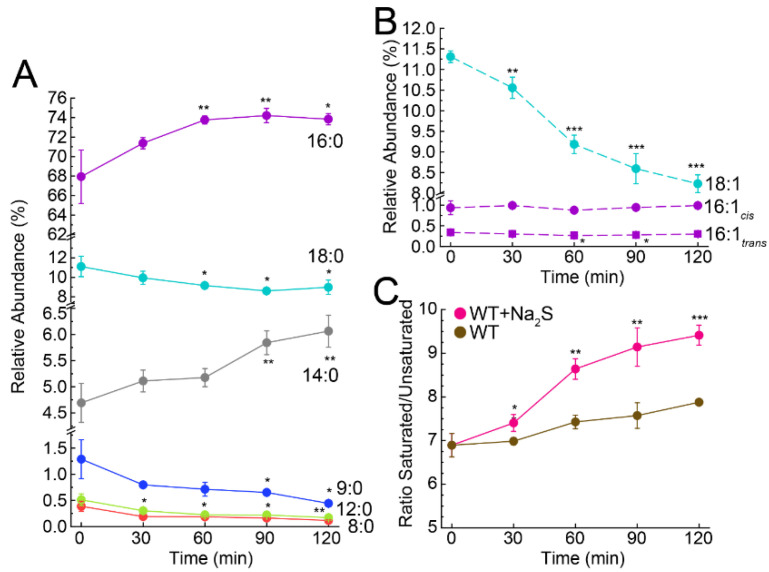
Fatty acid profiling in WT *E. faecalis*. Changes in the relative abundance for (**A**) saturated and (**B**) unsaturated fatty acids after the addition of exogenous Na_2_S to cell cultures. (**C**) The ratio of total saturated to unsaturated fatty acids for Na_2_S treated cells vs. untreated wild-type cells. Values represent the means ± S.D. derived from the results of biological triplicate experiments, with statistical significance established using a paired t-test relative to the endogenous, time 0 (*** *p* ≤ 0.001, ** *p* ≤ 0.01, * *p* ≤ 0.05).

**Figure 4 antioxidants-11-01607-f004:**
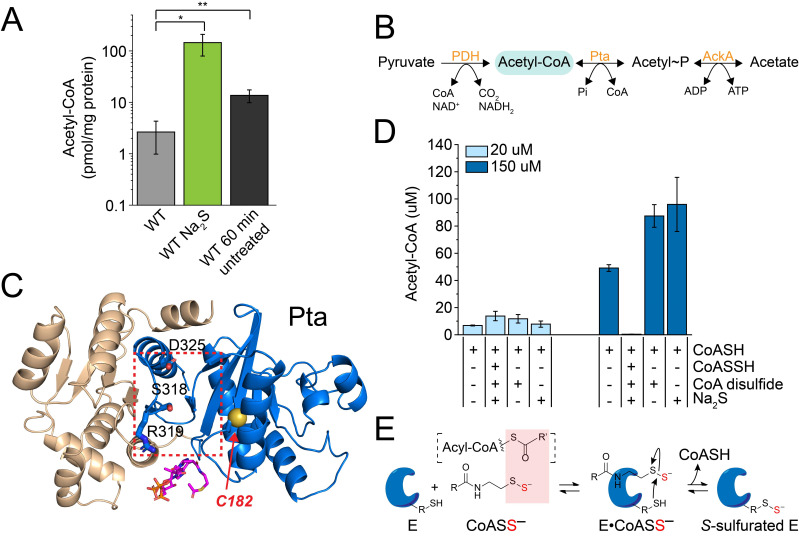
Acetyl-CoA accumulation via enzyme inhibition by CoASSH. (**A**) Quantitation of acetyl-CoA in WT cells treated with or without exogenous Na_2_S. Values represent the means ± S.D. derived from results of *n* = 5 replicates, with statistical significance established using a paired *t*-test to wild-type untreated cells (** *p* ≤ 0.01, * *p* ≤ 0.05). (**B**) Metabolic pathways for the biosynthesis of acetyl-CoA in *E. faecalis*. (**C**) Ribbon representation of an AlphaFold2 [67,68] model of *Ef*Pta. The cleft between domain I (wheat) and domain II (blue) defines the active site (red box), where R319 provided electrostatic stabilization to the nucleotide end of bound CoASH *(magenta*, from PDB 6IOX), S318 hydrogen-bonds to the terminal thiol and D325 is the catalytic base [69]. C182, with the sulfur atom shown as a yellow sphere, is extensively persulfidated in cells. (**D**) Impact of CoASSH on the yield of acetyl-CoA by *Ef*Pta. (**E**) Protein *S*-sulfuration via CoASSH poisoning. Binding of CoASSH to an acyl-CoA or CoA-utilizing enzyme, E, which has a nearby Cys that participates in persulfide transfer from CoASSH results in the release of the free thiol and a *S*-sulfurated enzyme.

**Figure 5 antioxidants-11-01607-f005:**
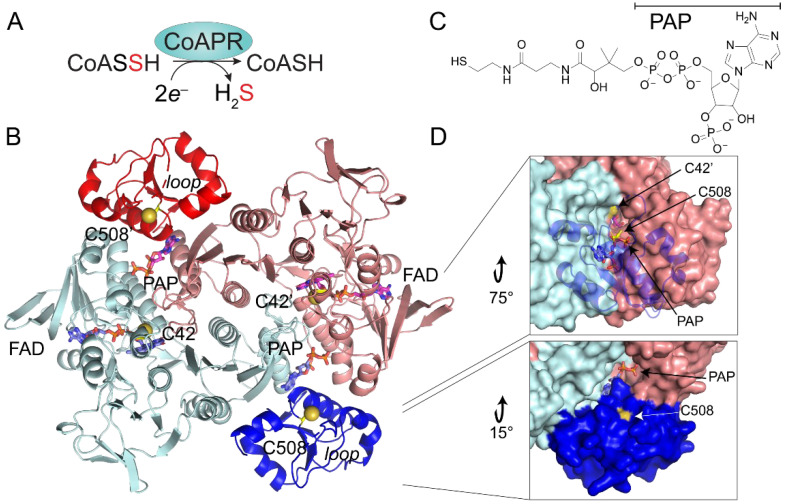
X-ray crystallographic analysis of *E. faecalis* CoAPR. (**A**) Reaction catalyzed by CoAPR; FAD and pyridine nucleotide cofactors are not shown. (**B**) Ribbon diagram of the CoAPR homodimer, with one subunit shaded light blue (CDR domain)/blue (sulfurtransferase domain) and the other subunit shaded salmon (CDR domain)/red (sulfurtransferase domain). FAD, PAP and the two catalytically required Cys residues (C42, C508) are shown in stick. Loop, residues E474-Q479. (**C**) Chemical structure of coenzyme A with the PAP subdomain indicated. (**D**) Two views of the surface representation of the CoAPR dimer, illustrating the surface accessibility of the C508 thiol from the diphosphate moiety of PAP (lower panel) and looking through the cartoon representation of the sulfurtransferase domain to see the tunnel that extends along the protomer interface from the PAP to C42 and the flavin.

**Figure 6 antioxidants-11-01607-f006:**
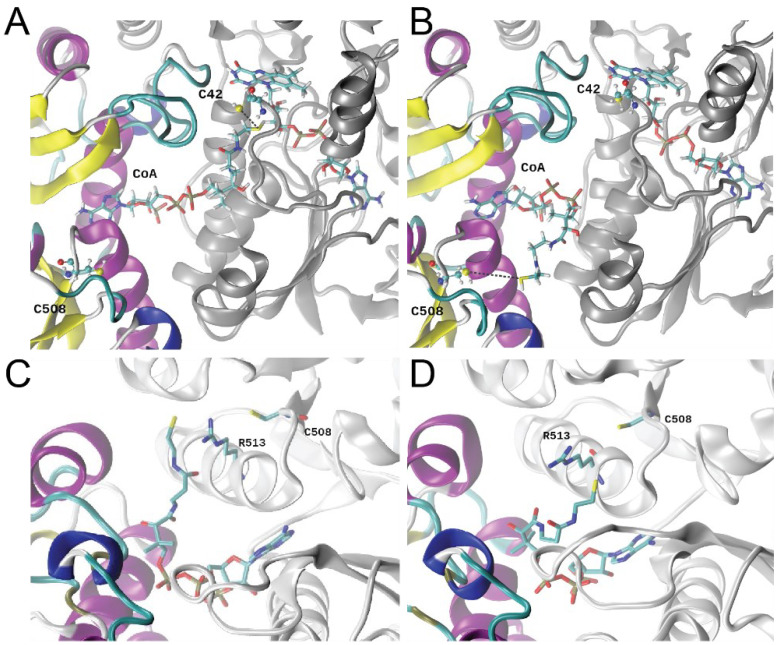
Representative snapshots of stable conformations of CoA relative to C508, obtained from 1.4 µs MD simulations. (**A**) Starting “extended” CoA conformation, protomer B; (**B**) representative “bent” CoA conformation, protomer B. The dimer backbone is depicted in the ribbon, with the CDR domain of one protomer colored based on a secondary structure and the rhodanese domain from the adjacent subunit shaded silver. All residues are depicted with sticks. The position of the pantothenate arm is slightly different in replica 1 (**C**) vs. replica 2 (**D**). approaching C508 from opposite sides of the R513 side chain (shown in sphere and stick).

## Data Availability

Crystallographic data have been deposited in the Protein Data Bank (http://www.wwpdb.org/), with the atomic coordinates and structure factors, under accession code 8A56. Models are available in ModelArchive (http://modelarchive.org/) with the accession codes ma-4xl3y (Pta), ma-eknu4 (FabN) and ma-df7nk (AccAD)_2_.

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
