# Peer review of "Metabolic and Structural Insights into Hydrogen Sulfide Mis-Regulation in Enterococcus faecalis"

_antioxidants, 2022, doi:10.3390/antiox11081607_

Round 1

Reviewer 1 Report

By employing exogenous sulfide, Walsh et al. show the proteomic response of E. faecalis in terms of H2S/RSS-mediated protein persulfidation. The proteomic analysis showed that significant changes in abundance of some proteins took place, mainly those affecting the biosynthesis of CoA and fatty acids, in response to persulfidation.

The paper is very well and clearly written, the experiments were properly conducted and the obtained results are relevant. The manuscript represents a significant contribution to the understanding of the underlying mechanisms of the effects of altered exposure of bateria to H2S/RSS species on their resistance to membrane damaging agents, such as antibiotics.

Reviewer 2 Report

The authors have examined various aspects of cell function related to H2S regulation in E faecalis. The specific methods appear to be sound but the study suffers from a diffuse and unfocused design. Rather than a systematic examination of a focused hypothesis, the report comes across as a collection of loosely related observations. For example, they show effects on Acetyl CoA and fatty acid regulation and conclude that these are connected, but there are no experiments to test such a connection. 

The methods are described in great detail. While this is helpful for new or unusual techniques, most are minor modifications of previously published procedures. Most of this description can be replaced by appropriate reference.

Round 2

Reviewer 2 Report

The authors have significantly condensed and focused the submission making it much more readable. It should serve as a useful body of information for investigators in the field.